# Mechanisms of Regulation in Intraflagellar Transport

**DOI:** 10.3390/cells11172737

**Published:** 2022-09-02

**Authors:** Wouter Mul, Aniruddha Mitra, Erwin J. G. Peterman

**Affiliations:** Department of Physics and Astronomy, and LaserLaB Amsterdam, Vrije Universiteit Amsterdam, 1081 HV Amsterdam, The Netherlands

**Keywords:** cilia, intraflagellar transport, axoneme, microtubules, motor proteins, kinesin-2, dynein, kinesin, motor regulation

## Abstract

Cilia are eukaryotic organelles essential for movement, signaling or sensing. Primary cilia act as antennae to sense a cell’s environment and are involved in a wide range of signaling pathways essential for development. Motile cilia drive cell locomotion or liquid flow around the cell. Proper functioning of both types of cilia requires a highly orchestrated bi-directional transport system, intraflagellar transport (IFT), which is driven by motor proteins, kinesin-2 and IFT dynein. In this review, we explore how IFT is regulated in cilia, focusing from three different perspectives on the issue. First, we reflect on how the motor track, the microtubule-based axoneme, affects IFT. Second, we focus on the motor proteins, considering the role motor action, cooperation and motor-train interaction plays in the regulation of IFT. Third, we discuss the role of kinases in the regulation of the motor proteins. Our goal is to provide mechanistic insights in IFT regulation in cilia and to suggest directions of future research.

## 1. Introduction

Our knowledge of cilia has developed substantially since they were first described as ‘incredibly thin feet’ by Anthony van Leeuwenhoek in the 17th century [1]. Cilia are organelles protruding from many eukaryotic cells, with a core consisting of a cylindrical microtubule (MT) structure, the axoneme. The axoneme is enveloped by a membrane that is a continuation of the plasma membrane, with, however, a distinct phospholipid composition [2]. Cilia are ancestral structures, present in all major groups of eukaryotes. There is, however, a substantial degree of specialization of cilia in different species and different tissues [3,4,5,6]. Generally, two different types are discerned: motile cilia (a.k.a. flagella) and primary cilia. Motile cilia drive the motility of the cell as a whole (e.g., the flagellum of a sperm cell) or of the liquid around the cell (e.g., the cilia of lung epithelium cells that move mucus). In contrast, primary cilia are static appendages, serving as antennae for signaling molecules or for environmental cues, for example repellent or attractant chemicals. In humans, malfunctioning cilia can lead to diseases, so-called ciliopathies. Ciliopathies come with a wide range of symptoms, most of them involving neurological disturbance [7,8]. In many cases, they are caused by genetic defects in ciliary genes. Equivalents of such mutations in model organisms have greatly contributed to our understanding of ciliary structure and function.

For assembly, maintenance and function, cilia depend on a specialized intracellular transport mechanism called intraflagellar transport (IFT). IFT was first described in 1993 by K.G. Kozminski et al. in *Chlamydomonas* flagella [9]. IFT, illustrated in Figure 1, is bidirectional: large protein complexes called IFT trains, consisting of IFT-A and IFT-B subcomplexes, move from ciliary base to tip, in the anterograde direction. At the tip they turn around to move in the retrograde direction back to the base [10,11]. IFT has been studied extensively using light and electron microscopy, and biochemical and genetic means, allowing the identification of the key proteins involved. Anterograde IFT is driven by heterotrimeric motor proteins of the kinesin-2 family, and in some species homodimeric kinesin-2 motors are also involved [12]. Retrograde transport is driven by cytoplasmic dynein-2, a.k.a. IFT dynein [13]. Motor activity appears to be tightly regulated in IFT, e.g., IFT dynein is transported by anterograde trains as inactive cargo [14]. Apart from motor proteins, cargoes that have been identified include structural ciliary components such as tubulin [15] and membrane proteins involved in sensing and signaling such as G-protein couple receptors [16,17].

Although cilia and IFT components are well-conserved over eukaryotes, important differences exist. To study cilia, a variety of model systems has been used, including unicellular protists (e.g., *Trypanosoma brucei*), *Xenopus leavis* and sea urchin [18]. Arguably, most of the detailed knowledge of structural and dynamical aspects of IFT and the motor proteins involved originates from research on three model systems: the flagella of *Chlamydomonas reinhardtii*, the chemosensory cilia of *Caenorhabditis elegans* and the primary cilia of cultured mammalian cells (Figure 2A). These model cilia have historically been explored primarily because of the relative ease with which genetic engineering, live fluorescence imaging and high-resolution electron microscopy (EM) can be performed. The recent progress in cryo-EM imaging with almost atomic resolution, in situ, in *Chlamydomonas* flagella has been particularly game changing [10,19]. Important differences have been identified between the cilia of these three different species (Figure 2B–D), including (i) the basal body at the ciliary base is missing in *C. elegans* [20], (iii) significant diversity in axoneme structure and composition exists between motile and primary cilia [21], (ii) in *C. elegans* cilia, two different kinesin-2 motors—one heterotrimeric, the other homodimeric– drive anterograde transport, while in many other cilia only a single heterotrimeric kinesin-2 is involved [12] and (iv) unlike in *C. elegans*, in *Chlamydomonas* kinesin-2 is not an inactive cargo associated with retrograde trains [22].

In the past decade, cilia and IFT have been extensively reviewed. Excellent general reviews have appeared [23,24], as well as reviews focusing on structural aspects [10,25], IFT motor proteins [12,26,27] and post-translational modifications in the cilium [28]. In this review, we will focus specifically on the regulation of IFT. As described above, during IFT, retrograde and anterograde motor activity appear tightly controlled, to make sure IFT trains travel from base to tip and back again in an orderly fashion. During ciliogenesis of *Chlamydomonas* cilia, IFT is upregulated and once the appropriate length is reached, IFT is downregulated again [29,30]. This tight regulation of IFT is also observed during cilium disassembly [31]. Recently, it has been shown that in *C. elegans* chemosensory cilia, IFT is affected by chemical stimulation. Chemical stimulation not only results in neuronal signaling, but also causes a considerable redistribution of IFT components over the cilium [32]. These observations show that IFT is indeed under tight control, and that this control is used by the cilia to adapt to different conditions. In this review, we will focus on IFT regulation from three perspectives: (i) from the perspective of the track, the axonemal MTs; (ii) from the perspective of the motor proteins involved in IFT; and (iii) from the perspective of the regulatory proteins such as kinases.

## 2. IFT Regulation from the Perspective of the Track

The core structural element of cilia is an axoneme, build of MTs, which serves as the track for the IFT motor proteins. In most cilia, the axoneme consists of nine doublet MTs in a cylindrical configuration, with their minus ends at the ciliary base emanating from the basal body, and their plus ends pointing towards the ciliary tip (Figure 2B). MT doublets are composed of a complete, cylindrical A-tubule build out of 13 protofilaments (PFs) and an incomplete B-tubule, parallel and connected to it, consisting of only 10 PFs [33,34]. In most primary cilia, the A-tubules extend further than the B-tubules, forming the so-called distal segment (DS) consisting of MT singlets (i.e., only consisting of a 13-protofilament A-tubule), beyond the proximal segment (PS) composed of MT doublets. For example, in mammalian olfactory cilia the DS can be as long as 100 μm [35], in *C. elegans* chemosensory cilia it is several micrometers long [36], in *Chlamydomonas* flagella it appears very short (few 100 s of nanometers) [37], and it is nonexistent in *Trypanosoma* motile cilia [38]. The axonemes of most motile cilia consist of an additional pair of singlet MTs in the center, in a so-called 9 + 2 configuration (in contrast to the 9 + 0 configuration of sensory cilia). Motile cilia contain, in addition, the complete machinery (including central apparatus, radial spokes, nexin links and axonemal dynein) for flagellar beating [19]. In recent years, impressive progress has been made in resolving the structure of this machinery in *Chlamydomonas* flagella, close to the atomic level, using cryo-EM [39,40,41,42]. These new structures reveal a very dense, complex meshwork of proteins, including many proteins that are bound to the MTs: MT-inner proteins (MIPs) and outer MT-associated proteins (MAPs) (Figure 3). So far, this level of structural detail has not been obtained for sensory cilia.

In the remainder of this section of our review we want to focus on how the axonemal MT tracks regulate IFT-motor function. We will focus in depth on what happens at the base of the cilium, where IFT trains assemble and take off for their anterograde journey towards the tip. We will then move our focus to the ciliary tip, where, after arrival, trains partly disassemble and reassemble into retrograde-moving trains. Finally, we will focus on the specific tracks (microtubules or even protofilaments) IFT trains might take. Firstly, however, we will briefly address the emerging picture on the role played by the specific ‘tubulin code’ of the microtubules in motor and IFT regulation in cilia.

### 2.1. The Tubulin Code

Over the past decades, the concept of the tubulin code has emerged, which poses that the diversity in function and properties of MTs finds its origin in a combination of distinct tubulin isotypes and post-translational modifications (PTMs, including polyglutamylation, polyglycylation, acetylation and detyrosination; Figure 3) [43,44,45]. These two can directly affect MT properties (including stability, mechanical properties, and the interaction with proteins) or indirectly by modulating interactions with MAPs. Of particular interest to this review is the observation in in vitro motility experiments that the activities of cytoplasmic dynein and kinesins (including ciliary kinesin-2) are substantially affected by tubulin isoforms and PTMs [46]. It is well-established that cilia make use of a specific subset of tubulin isoforms. For example, *C. elegans* contains genes for 9 α- and 6 β-tubulins, of which only 3 (TBA-5, TBA-6, TBA-9) and 1 (TBB-4), respectively, appear to be present in the chemosensory cilia [47,48,49]. It is also evident that PTMs play important roles in ciliary microtubules. In *C. elegans* chemosensory cilia, polyglutamylation of the C-terminal tails of α- and β-tubulin by TTL-4, TTL-5 and TTL-11 and deglutamylation by CCP-1 are the prevalent PTMs [47]. Polyglutamylation has been shown to have an effect on IFT velocity [50] and location of the motor proteins in the cilia [51]. Additionally, in *Chlamydomonas* there is evidence that primarily the B-tubule is polyglutamylated [52]. Up to now it is not fully understood how tubulin PTMs affect IFT. Further quantitative analysis of the dynamics of IFT and motor parameters might provide deeper insights.

### 2.2. Polarity of the Axoneme and Directionality of IFT Trains

The ciliary axoneme is polar: the minus ends of all the constituting MTs are located at the ciliary base, and the plus ends at the tip. This is crucial for the directionality of IFT, since kinesin-2 motors are plus-end directed, while IFT dynein is minus-end directed [26]. A specific property of IFT is that trains appear to move in one go, from one extremity of the cilium to the other. In other transport systems, e.g., axonal transport of vesicles and organelles, often signatures of ‘tug of war’ are observed: membrane-enveloped cargoes contain different motor proteins that pull the cargo in opposite directions [53]. This can result in one team of motors ‘winning’ for a while, causing a bout of unidirectional transport. At random moments or locations, the opposite direction team can increase its grip on cargo and MT, resulting in the reversal of transport direction or a temporal stall. Signatures of such a tug of war have not been observed for IFT, suggesting that motor activity is under tight control: an IFT train is either an anterograde-moving train, with kinesins active and IFT dynein in an inhibited state, or the opposite. Switching between these two states should then happen specifically at the tip and base.

### 2.3. IFT-Train Turnaround at the Ciliary Tip

Quantitative imaging of IFT components changing direction at the tips of *Chlamydomonas* flagella [22,54] and *C. elegans* chemosensory cilia [55] has revealed that anterograde IFT trains disassemble, at least partially, upon arrival at the tip (Figure 4A). After subsequent reassembly, retrograde trains take off towards the base. This turnaround process appears to take less than a few seconds. It has been speculated that the turnarounds require specific proteins that have been shown to be enriched at the tip [56,57] and might involve kinases [58]. *C. elegans* chemosensory cilia that lack OSM-3 function (one of the two kinesin motors in this system) and consequently the distal MT singlets at the tip, however, do not show substantially altered turnaround behavior at their truncated tips [55]. In a recent study in *Chlamydomonas* it was shown that a physical barrier, a wedge blocking IFT at any location along the cilium, can trigger turnarounds from anterograde to retrograde directions, indicating that it is the end of the MT track that triggers turnaround and not specific proteins at the ciliary tip [37]. Furthermore, in EM images of a partly ‘derailed’ IFT train at the tip, the part of the train still in contact with the MT track shows the typical, highly ordered anterograde-train structure. However, the other part of the train that has run off the track becomes disordered, suggesting that train conformation is bistable and dependent on interaction with the axonemal track. Retrograde-moving trains, in contrast, accumulate at the wedge. This is not unexpected, since in *Chlamydomonas*, kinesin motors are not cargo of retrograde IFT trains [22], in contrast to in other cilia. Furthermore, it has been hypothesized that for the retrograde-to-anterograde conversion, trains need to be completely reassembled at the ciliary base. It would be interesting to perform such physical IFT-blocking experiments in other ciliary systems, such as *C. elegans*. This might, however, be complicated because the chemosensory cilia in *C. elegans* are inside the phasmid or amphid channels within its body. 

### 2.4. IFT-Train Assembly at the Ciliary Base

For cilia in several organisms, it has been shown that a pool of IFT components is maintained at the ciliary base [59,60,61,62]. This pool consists partly of proteins returning from their IFT journey from the ciliary tip, but is also replenished from the cytosol. The extent (and time scale) of this replenishment might differ between cilia. Fluorescence recovery after photobleaching (FRAP) experiments on *Chlamydomonas*, *Tetrahymena thermophila*, mammalian IMCD3 cells [62], and *Xenopus* multiciliate cells [60] have indicated that the exchange between basal pool and cytosol is mostly open. This is in contrast to *Trypanosoma*, where the exchange between basal pool and cilium as well as cytosol appears less substantial [59]. It is unclear whether these different observations point to actual biological differences or are due to experimental differences (e.g., probing at different times scales). It has, however, been convincingly shown that FRAP time scales for different IFT proteins are distinct, which suggests that IFT components assemble in a sequential way [60,62]. Further evidence for this comes from a recent study combining super-resolution fluorescence imaging and cryo-EM [63]. In the EM structures, in situ assembling IFT trains can be discerned at single-protein resolution. The structures indicate that IFT-B complexes form first, followed by attachment of IFT-A, and finally IFT dynein (Figure 4B). In the EM images kinesin-2 could not be discerned, but super-resolution fluorescence imaging indicated that kinesin-2 is the last to attach to assembling IFT trains before they take off on their anterograde journey. It would be interesting to see whether this sequential assembly of IFT trains in *Chlamydomonas* is a general mechanism in cilia from other organisms. An important aspect in this respect would be the attachment of kinesin-2 to assembling trains, since this motor diffuses from tip to base in *Chlamydomonas* [22], while it is actively transported back by retrograde trains in *C. elegans* [61]; Figure 2D). Furthermore, in *Chlamydomonas*, the ciliary base appears to be in open contact with the cytosol, while this might be different in other systems (e.g., in *C. elegans* chemosensory neurons, cilia are attached to the neuron’s dendrite with the so-called periciliary membrane compartment (PCMC) that appears to have a different composition than cilium and dendrite [64]).

### 2.5. Track Specificity of IFT Trains

The EM structures of *Chlamydomonas* flagella discussed above [39,40,41,42] show that cilia, in particular motile ones, are very crowded. There appears to be only very limited space left for moving IFT trains, which raises the question how trains moving in opposing directions could pass. Correlated fluorescence and electron microscopy in *Chlamydomonas* has demonstrated that trains do not collide, but that anterograde and retrograde trains use different track on the MT doublets: anterograde trains use the B-tubules, while retrograde trains use the A-tubules [65] (Figure 4C). It is not known what the molecular basis is of this preference of the different motors for these different tracks. The MT lattice in the A- and B-tubules is the same [66], but it has been reported that the MT-isoform composition and PTMs might be different [67]. IFT kinesins have shown to not only step in the forward direction along a MT, but also to make sidesteps, causing some motors to move along MTs in a left-hand helical path in vitro [68]. On axonemes, however, no helical paths were observed, which could be explained by the motors hitting a steric barrier at the connection between A- and B-tubules and as a consequence follow a straight trajectory along a single PF. Sidestepping has also been observed in *C. elegans* chemosensory cilia [69]. It is, however, unclear whether tubule preference is unique to *Chlamydomonas*, or universal and similar in other organisms. EM of mammalian primary cilia appears to show that anterograde IFT trains move along A-tubules in the singlet region [70], but retrograde trains were not resolved.

In *Trypanosoma* a different level of track specificity has been observed. In this organism, IFT only appears to take place along a small subset of MT doublets [71], with anterograde and retrograde IFT taking place on different sides of the axoneme. Such MT specificity within the axoneme does not appear to be the case in *Chlamydomonas* [63] and *Xenopus* multiciliate cells [60], where all 9 MT doublets appear to be occupied by loading IFT trains. It might be that the specific anatomy of the *Trypanosoma* cilium (which is attached to the cell body via a paraflagellar-rod structure facing only part of the MTs in the axoneme) breaks the (almost) cylindrical symmetry observed in other cilia, resulting in functional differentiation of the MT doublets forming the axoneme. It might also be that specific MT PTMs are involved. Further high-resolution structural and functional imaging of cilia from different organisms will be required to get further insights.

## 3. IFT Regulation from the Perspective of the Motors

Efficient cycling of IFT trains in cilia requires intricate regulation of motor activity. Recently, IFT and IFT motors have been described from a mechanistic and structural viewpoint in excellent reviews [10,12,23,25,27,72]. In this part of our current review, we will focus on the different aspects of motor action, cooperation and regulation that coordinate the efficient cycling of IFT trains in the cilia. First, we will summarize the role, structure, motility properties and possible inhibition mechanisms of the key IFT motors: heterotrimeric kinesin-2, homodimeric kinesin-2 and IFT dynein. Then we will discuss the interaction of IFT motors with IFT trains. This will be followed by considerations on (i) IFT-traffic management at the TZ, (ii) collective motor action, (iii) dynamicity of the motor-train interaction, and (iv) role of axoneme tip dynamics on IFT. 

### 3.1. The Properties of IFT Motors

Heterotrimeric kinesin-2. Anterograde IFT is driven primarily—exclusively in several organisms—by heterotrimeric kinesin-2. The special feature of this kinesin is that it consists of two non-identical motor subunits, 2α and 2β (FLA8 and FLA10 in *Chlamydomonas*; KLP11 and KLP20 in *C. elegans*; KIF3A and KIF3B in mammals), and a non-motor subunit, KAP (FLA3 in *Chlamydomonas*; KAP1 in *C. elegans*; KAP3 in mammals), see Figure 1 [12]. To understand the role of the different subunits and why kinesin-2 has evolved to be heterotrimeric has been an important direction of research. In in vivo studies, the heterotrimeric organization has been shown to be essential, since deletion mutants of all three subunits disrupt IFT [73,74,75,76]. The motor subunits contain an N-terminal motor domain, a neck domain, a charge-rich hinge region (which prevents homodimerization [77]), a coiled-coil stalk (which facilitates heterodimerization [78]) and a disordered tail region (which is potentially a key region for regulation, with several phosphorylation sites [25,58,74,79]). The KAP subunit preferentially binds and potentially stabilizes the coiled-coil stalk of the heterodimer [80] and has been speculated to play a regulatory role in the activity of kinesin-2 [12]. While homodimerization of the individual motor subunits may be possible in vitro [81] or by overexpression [82], heterodimerization is strongly favored when both motor subunits are present. KAP is required for targeting of kinesin-2 to the ciliary base [83] and it binds preferentially to the 2β motor subunit. The 2α motor subunit interacts with anterograde IFT train proteins [82], indicating that the functional roles of the heterotrimer are distributed over the subunits. In vitro studies have been performed to probe the properties of chimeric kinesin-2 constructs with homodimeric motor domains. These studies suggest that the 2α and 2β motor domains have different velocities, processivities and load-bearing properties [84,85,86]. Ιn vivo, heterotrimeric kinesin-2 velocity varies significantly between organisms: ~2.2 µm/s in *Chlamydomonas* and *Trypanosoma* [62,71,87] and ~0.5 µm/s in *C. elegans* and mammalian cells [61,88,89]. Swapping motor domains between kinesin-2 of different species has revealed that motor velocity is intrinsic to the motor domains [90]. In *Chlamydomonas*, swapping the motor domains for the slower ones from mammals indeed slows down IFT, which has an impact on axoneme length [91]. In vitro studies have indicated that kinesin-2 can adopt an autoinhibited state, like other kinesins [92], with the tail or the coiled-coil region folding onto the motor domains [84]. Autoinhibition can be relieved by binding the motor to a bead [86] or surface [84], or by specific mutations that prevents the heterodimer from folding back [84]. It has been hypothesized that, in vivo, autoinhibition is relieved by binding of the tail domains to IFT trains, with the KAP subunit playing a crucial role [90].

Homodimeric kinesin-2. In contrast to *Chlamydomonas*, in *C. elegans*, anterograde IFT is driven by two distinct kinesin-2 motors: heterotrimeric kinesin-II and homodimeric OSM-3 [73]. OSM-3 has two identical motor subunits (2γ; Figure 1), is processive and faster than kinesin-II (~1.3 µm/s versus ~0.5 µm/s) [73,93]. In recent years, it has been discovered that kinesin-II is responsible for carrying the anterograde trains from the base across the transition zone, after which kinesin-II gradually falls off the trains in the ‘handover zone’ and is replaced by OSM-3, which carries the trains to the tip at a higher speed [61] (Figure 4F). Remarkably, mutant worms lacking kinesin-II function still contain cilia that appear fully grown and functional, although the ciliary structure is altered and the rate of injection of anterograde IFT trains is lower [61,94]. This suggests that the two kinesin-2 s in *C. elegans* are, to some degree, functionally redundant. On the other hand, the axonemes in cilia of mutant *C. elegans* strains lacking OSM-3 function are much shorter and the DS is missing, which renders the cilia functionally defective. Anterograde IFT is slower in these worms, since it is driven only by kinesin-II [61,93]. It is possible that this leads to a slower rate of supply of tubulin (cargo of anterograde IFT trains [48,95,96,97]) to the axoneme tip, causing the axoneme to be shorter. Mammalian cells also contain a homodimeric kinesin-2, KIF17, which also localizes to cilia [35]. Interestingly, a recent study indicates that KIF17 does not participate in driving anterograde IFT, but interacts with IFT trains as passive cargo [35,89,98,99]. Both OSM-3 and KIF17 have a similar autoinhibition mechanism as heterotrimeric kinesin-2, with the folded conformation of the tail and coiled-coil domain interacting with the motor domain, which can be relieved by similar mutations [100,101]. Furthermore, Dyf-1 (IFT-70), part of IFT-B complexes, has been shown to directly relieve the autoinhibition of OSM-3 in *C. elegans* [102]. The role and mechanism of activation of KIF17 in mammalian cilia has remained unclear, although KIF17 has been suggested to play a role in cycling channel proteins [103,104]. 

In summary, the kinesin-2 family of kinesin motor proteins still hold many unanswered questions. Why are several members heterotrimeric? Most kinesins are homodimeric [105], with a pair of identical motor subunits, often using adaptor proteins to connect to cargo [106]. Furthermore, the phylogeny of kinesin-2 motors is peculiar, involving separate and unrelated gene duplication events of a putative common ancestral homodimeric kinesin-2γ gene. This results in *Chlamydomonas* heterotrimeric kinesin-2 being more closely related to *C. elegans* OSM-3 (homodimeric) than kinesin-II (heterotrimeric) [21]. Finally, in some organisms (*Chlamydomonas*) one kinesin-2 motor is sufficient to drive anterograde IFT, in others (*C. elegans*) two are employed, while for a third class of cilia it is not fully understood whether and how a second motor is involved (KIF17 in mammalian primary cilia) [37,107]. It will be fascinating to find out whether this is due to differences in the motor proteins (for example from the regulatory point of view) or intrinsic to the differences in structure, size and function of the different cilia.

IFT dynein (cytoplasmic dynein-2). Retrograde IFT is driven by IFT dynein (a.k.a. cytoplasmic dynein-2). Recent advances in EM have contributed to a better (structural) understanding of the motor, as recently reviewed [10,25,27,108]. IFT dynein is closely related to cytoplasmic dynein-1, the main MT-based, minus-end directed motor active in many cellular processes outside the cilium [109]. IFT dynein, in contrast, appears to be active only in cilia [110,111,112] and in an autoinhibited state elsewhere. IFT dynein is a large (~1.4 MDa), multi-subunit complex, with a homodimer of IFT-dynein heavy chains (DHC2) at its core. The heavy chains contain a C-terminal motor head, composed of a AAA+ motor domains, a MT-binding stalk, and an N-terminal tail domain involved in dimerization [108]. The tail interacts with two copies of IFT-dynein light-intermediate chain (LIC3) and two dissimilar intermediate chains, WDR34 and WDR60, which form a highly asymmetric heterodimer with their N-terminal regions held together by several light-chain dimers (Figure 1) [113,114]. Recent high-resolution EM structures of anterograde IFT trains in *Chlamydomonas* flagella have shown that IFT dynein is tightly stacked onto the trains in an autoinhibited conformation, with the MT-binding stalks pointing away from the axoneme [14,114]. At the ciliary tip, trains appear to undergo a substantial structural change, with activated dynein engaging with newly formed retrograde trains in an activated state. At this point, no in situ high-resolution structures of activated IFT dynein attached to retrograde IFT trains are available. Consequently, the molecular basis for IFT-dynein activation is unknown. It is important to note that dynactin and adaptor proteins—well-documented activators of cytoplasmic dynein-1 [109]—are not present in cilia. It could be that the activation of autoinhibited IFT dynein is linked with conformational changes upon detachment from anterograde trains and re-association to converted retrograde trains. Little is, however, known about the regulatory machinery that controls this conformational change.

### 3.2. Interactions of IFT Motors with IFT Trains

As discussed above, recent cryo-EM studies in *Chlamydomonas* and mammalian cilia have revealed detailed new insights in the structure of anterograde IFT trains [14,70,114]. Anterograde trains are densely packed and ordered polymeric structures (>80 MDa), on average ~312 nm long (in *Chlamydomonas*) and ~50 nm wide, with periodic repeats of IFT-B, IFT-A and IFT-dynein protein complexes of approximately 6, 11 and 18 nm, respectively (Figure 4E). Based on these structures, it was suggested that IFT-B forms the core of the train to which IFT-A and IFT dynein attach. IFT-A and IFT dynein’s MT-binding stalk domain are positioned on the membrane side, away from the axoneme [14]. The periodic repeats suggest a stoichiometry of 2:6:3 for IFT-dynein:IFT-B:IFT-A, but IFT-A and IFT dynein were lacking on the ends of the IFT-B core structure, resulting in an overall stoichiometry of 2:8:4 in trains [14]. 

Genetic and biochemical studies have indicated that heterotrimeric kinesin-2 from *Chlamydomonas* and mammals interacts with IFT-B proteins IFT52, IFT88, IFT38 and IFT57 [82] that together form the connection between the IFT-B1 (core) and IFT-B2 (non-core) sub-complexes. Furthermore, another study has found that heterotrimeric kinesin-2 interacts with IFT-B2 protein IFT54 [115]. Surprisingly, in *C. elegans* it has been proposed that heterotrimeric kinesin-II binds to the IFT-A complex, although this is based on indirect evidence [93,116]. If this were to be the case, then it would suggest that the interaction between heterotrimeric kinesin-2 and IFT trains as well as the train structure is radically different in *C. elegans* compared to *Chlamydomonas* (where EM images indicate that IFT-A is not close to the axoneme but on the membrane side of trains). Future studies, for example including in vitro reconstitution of IFT trains [102], or cryo-EM will be required to resolve these contrasting observations. Another remarkable difference between cilia from different species is that in *Chlamydomonas* heterotrimeric kinesin-2 diffuses back to the base in an inactivated form [22], while in *C. elegans* this anterograde motor is transported back as cargo associated to retrograde trains. It is not known to what protein (domain) in retrograde IFT trains kinesin-II binds. The retrograde fate of heterotrimeric kinesin-2 in mammalian cilia is unclear, and single-molecule explorations in such systems will be needed to shed light on this. In *C. elegans*, activated homodimeric OSM-3 is understood to associate with anterograde IFT trains via IFT70 (Dyf-1 in *C. elegans*) [102]. In a recent in vitro reconstitution study, it has been shown that OSM-3 attaches, in the autoinhibited state, to the IFT46 and IFT56 subunits of, most likely, retrograde IFT trains [117]. Remarkably, it appears that activated mammalian KIF17 is not able to interact with IFT70, while autoinhibited KIF17 can interact with IFT46/IFT56 as passive cargo, just like OSM-3 [117]. 

The interaction of autoinhibited IFT dynein with anterograde IFT trains has been structurally resolved in recent cryo-EM studies. IFT dynein is tightly packed with neighboring IFT dyneins along the IFT-B template, each IFT dynein spreading over 7–8 IFT-B repeats, with many of the IFT-dynein subunits involved in interactions [27]. Further studies will be needed to identify the exact binding partners within IFT trains, although it is understood that IFT172 is an important player [118]. In contrast, cryo-EM studies of retrograde trains have not yet yielded similar quality structures, leaving many questions on the interaction of active IFT dynein with retrograde trains to be answered. Retrograde IFT trains appear as elongated, zig-zag-shaped structures with a periodic repeat of 43 nm. IFT dynein binds approximately every second repeat, resulting in a spacing of ~80–90 nm [14]. 

### 3.3. Trafficking across the Transition Zone 

At the ciliary base, the transition zone (TZ) and transition fibers form a physical barrier between the cell body and the cilia, which ensures that membrane proteins and large soluble proteins cannot freely diffuse between cilium and cytoplasm [119]. In the TZ, the first micrometer or so of the cilium, Y-shaped protein complexes link the axoneme to the ciliary membrane, forming a densely packed region [120]. For proteins to enter and exit the cilia, they have to associate to IFT-dynein or kinesin-2 driven IFT trains as cargo. Another ‘escape strategy’, which we will not further discuss here, is the shedding of membrane-enclosed vesicles from the cilia tip [47,121,122]. Fluorescence images of IFT trains along cilia indicate that IFT trains que up at the ciliary base before the TZ, suggesting that the MT lattice is very crowded in this region, which could affect the motility parameters of IFT motors [61,73,123], as illustrated in Figure 4D. Several kinesins, including kinesin-2, have been shown to be capable of switching protofilament lanes, which allows them to circumnavigate proteins acting as roadblocks [85,124,125,126]. In vitro motility studies have shown that *C. elegans* heterotrimeric kinesin-II is more efficient at sidestepping than homodimeric OSM-3 [69,124] and that its motility parameters are far less impacted by crowding [127]. This could explain the functional differentiation of the two kinesin-2 motors in *C. elegans*, where the slower kinesin-II drives anterograde IFT trains from the ciliary base across the TZ, after which the faster OSM-3 takes over the IFT trains to further drive transport towards the ciliary tip [61]. Similar in vitro investigations have not been performed for IFT dynein, but cytoplasmic dynein-1 has been shown to be capable of switching protofilament lanes [128,129] and avoid roadblocks [130]. Furthermore, its two MT-binding stalk domains have been shown to walk on different protofilaments (or MTs) [131,132]. It is possible that IFT dynein behaves similarly, allowing it to be an efficient retrograde motor, also in crowded regions such as the TZ and ciliary base.

In live *C. elegans*, anterograde and retrograde IFT velocities have been shown to be lower in the densely packed TZ than in the rest of the cilium [61]. In mutant worms lacking a functional TZ, this decrease in velocity close to the ciliary base was partially lifted [61]. In the TZ, IFT motors have to drag IFT trains past the Y-shaped linkers, which might result in substantial drag forces. The load-bearing properties of heterotrimeric kinesin-2 have been explored using optical tweezers in vitro [84,86,133,134]. Single kinesin-2 motors can exert forces up to ∼5 pN [134], similar to kinesin-1. Furthermore, resisting forces cause the motor to back-step and slip along the microtubule lattice [86], which might be advantageous when encountering a roadblock. Resisting loads hardly affects the velocity of the mammalian heterotrimeric kinesin-2 KIF3A/B, while the reduced length of processive runs indicates that resisting forces enhance detachment from the MT track [86]. It could be possible that in the TZ in vivo, kinesin-2 motors attached to an anterograde moving train are constantly detaching from the MT, thereby lowering the overall velocity. Since IFT trains are οccupied by several tens of motor proteins [61], almost always several remain bound to the MT track, as estimated from a force study in *Chlamydomonas* [135]. To understand the mechanism in detail, studies of the load-bearing properties of well-defined teams of kinesin-2 motors will be needed. As far as we are aware, no studies have been performed to explore the load-bearing properties of IFT dynein. 

### 3.4. Collective Motor Behavior

While several studies have looked into the collective behavior of MT-based motor proteins [136,137,138], no such studies have been performed for IFT motors. Furthermore, IFT trains are rigid, membrane-less structures that have evenly spaced motor binding sites. This is in contrast to the cargoes of most cytoplasmic, which are often membrane enveloped structures like vesicles or organelles. On anterograde IFT trains, activated kinesin-2 motors can bind every 6 nm, which does not match the 8 nm periodicity of tubulin dimers along a microtubule protofilament [10] (Figure 4E). This mismatch in periodicity could imply that either (i) kinesin-2 motors do not associate with every binding site on an IFT train, (ii) at any given point only a few kinesin-2 motors engage with the MT lattice or (iii) the motors use multiple protofilament lanes to transport an IFT train, with adjacent kinesin-2 motors stepping on different protofilaments. Activated IFT dynein associate with retrograde trains every 80–90 nm [14]. A recent study in *C. elegans* primary cilia suggests that when the amount of IFT dynein recruited for retrograde IFT is low, retrograde trains move erratically and IFT dynein cannot power retrograde IFT trains across the TZ [139]. While it is evident that the collective action of motors is crucial for robust IFT, only a few studies have addressed this aspect [90,140]. Considering the periodic geometry of IFT trains, it would be insightful to perform in vitro studies, involving attachment of IFT motors to DNA origami structures where parameters such as spacing, number and type of motor as well as rigidity of the template DNA structure can be readily altered [141].

### 3.5. Dynamic Interactions of IFT Motors with Trains

Another aspect that requires a closer inspection are the binding and unbinding kinetics of IFT motors to and from IFT trains (Figure 5). A recent single-molecule study in *C. elegans* has revealed that the interaction of IFT motors with IFT trains is highly dynamic [142]. Heterotrimeric kinesin-II motors were shown to stochastically detach from anterograde trains, switching from a directed to a diffusive state (likely autoinhibited). After a brief diffusive search, these motors reattached to another anterograde or retrograde train they encountered in their paths. The further an anterograde train has moved into the cilium, the lower the amount of kinesin-II still bound, resulting in kinesin-II cycling between the base and the beginning of the proximal segment. OSM-3 also shows directional switches, but at different locations. It cycles mostly between the beginning of the proximal segment and the tip. In this way, faster OSM-3 gradually replaces slower kinesin-II on an anterograde-moving train in the ‘handover zone’, resulting in a gradual velocity increase of the IFT train [61] (Figure 4F). It might be possible that the attachment of kinesin-II and OSM-3 to an IFT-B complex is mutually exclusive, with OSM-3 only capable of attaching to an IFT-B repeat when no kinesin-2 is bound. Indeed, kinesin-II-mutant *C. elegans* strains show a significantly higher density of OSM-3 motors at the ciliary base [61], suggesting that OSM-3 is capable of associating to and driving IFT trains at the base in the absence of kinesin-II. For *C. elegans* IFT dynein, it was observed that most individual motors fall off anterograde trains near the tip [55], but that some detach along the way throughout the cilium [143]. In *Chlamydomonas* and mammalian cells, dynamic interactions of heterotrimeric kinesin-2 and IFT dynein with IFT trains have not been reported. More detailed single-molecule studies, in conjugation with in silico approaches, might be required to improve our understanding on how motor-train interaction kinetics orchestrate IFT. It is also likely that kinases play a crucial role in modulating the attachment and detachment rates of the motors-train interaction along the length of the cilium, as discussed later.

### 3.6. IFT and Variations in Axoneme Length

It is well established that there is a strong link between IFT and ciliary structure, in particular length. On the one hand, mutations that block IFT (including deletion mutants of IFT motors) often cause defects in ciliogenesis [73]. On the other hand, tubulin, a key subunit of the axoneme, is an IFT cargo [48,95,96]. IFT and its track are thus intertwined in an intriguing way: the IFT machinery is deeply involved in building the track it moves along, by delivering the building blocks to the end of the track. It is likely that several MT plus-end binding proteins, potentially involved in regulating the dynamics and stability of the singlet and doublet tips of axoneme [144], are transported by IFT, regulating their local activity. For instance, in mammalian cells, the non-motile kinesin-4, Kif7, a passive cargo of IFT trains, localizes and stabilizes axonemal tips [145,146]. Ciliary length control by IFT has been investigated and modelled in detail in *Chlamydomonas*. The latest models indicate that key factors determining the length of the axoneme are the velocity of anterograde IFT, driven by kinesin-2 and the time it takes for kinesin-2 to diffuse back from tip to base [91,147]. In mammalian primary cilia, the small-molecule inhibitor of cytoplasmic dyneins, ciliobrevin, has been shown to cause defect in ciliogenesis, resulting in very short cilia [148]. Furthermore, acute chemical inhibition of genetically modified inhibitable heterotrimeric kinesin-2 led to complete loss of IFT, resulting cilium disassembly [89]. In *C. elegans* chemosensory cilia, fluorescence recovery after photobleaching experiments and modelling have shown that new tubulin, transported by IFT is incorporated in the axoneme at the ciliary tip and at the end of the proximal segment, where the B-tubules end [48]. A study employing dendritic laser ablation to block the supply of new material into cilia showed that IFT components redistribute after ablation to the base, followed by shortening of the axoneme [149]. More recently, similar behavior was observed in response to the chemical repellents the chemosensory cilia sense [32]. In these experiments, MTs shortened, disappearing from the distal segment in a reversible fashion, (reversibly) decreasing the ability of the cilium to react to subsequent stimuli. It is not clear what the underlying regulatory mechanism is to these changes that directly connect regulation of IFT to the regulation of ciliary structure and function. Furthermore, it is unknown whether a similar regulatory connection between IFT and ciliary structure is also present in other ciliary systems.

## 4. IFT Regulation from the Perspective of Regulatory Proteins

Several regulatory proteins have been linked to IFT regulation. We have briefly discussed tubulin PTM enzymes and their possible roles in IFT regulation. BBSome-complex proteins and tubby family proteins have been shown to regulate the connection between IFT and membrane-bound cargos [150,151]. It is unclear whether or how these proteins regulate IFT itself. Among other regulatory proteins that have been identified to play key roles in ciliary development, maintenance and function, most are kinases. In many cases these kinases also appear to affect IFT, but it is often unclear whether this is a direct effect and how the properties of IFT components are affected. In the following part of this review, we will give an overview of four different kinase families that play roles in IFT regulation in different organisms: RCK, CCRK, CaMK and CDKL (see Table 1 for names in different species). At this point in time, it appears that we are starting to understand what the effect is of individual regulatory proteins. How the whole regulatory network functions in regulating IFT remains, however, largely unknown.

### 4.1. RCK

One of the best studied kinase families involved in IFT regulation is the ros cross-hybridizing (RCK) kinase family—part of the group of CMGC kinases [171]. RCK members (Table 1)—CILK1 in human (cyclogenesis associated kinase 1, recently renamed from ICK) [153,172], LF4 in *Chlamydomonas* [158], DYF-5 in *C. elegans* [156,173]—have been shown to affect ciliary length. Knockouts result in longer cilia, with a larger variation in length and unusual accumulations of IFT components [156,158,174,175], for example at the tip. In mammalian cells, these tip accumulations have been shown to shed as extracellular vesicles [11,58,88,176]. In all three organisms, RCK members have been shown to associate with IFT [11,58,88,156,158,175,176]. The ciliary localization is, however, not the same: in *Chlamydomonas*, LF4 is enriched at the base [158,174,177], while in *C. elegans* and mammals, DYF-5 and CILK1 are enriched at the tip [11,156,175]. In *Chlamydomonas*, LF4 has been shown to be activated through phosphorylation by the length regulation complex (LRC), consisting of LF1, 2 and 3 [161,177]. A key phosphorylation target of RCK is heterotrimeric kinesin-2: in mammals KIF3A [58,79] and in *Chlamydomonas* FLA8, the homolog of KIF3B [177]. In mice, KIF3A has been shown to be not the only target of CILK1, since mutants expressing a non-phosphorylatable KIF3A variant have a distinct phenotype from CILK1-knockouts [178]. The phenotypes of the RCK-knockouts in different organisms–(tip) accumulation of IFT components– resemble those of dynein-2 and IFT-A knockouts [11,179,180,181,182]. This, together with the presumed tip localization of RCK, suggests that these kinases could play a role in regulating the turnaround of anterograde trains. At the tip, RCK could promote detachment of heterotrimeric kinesin-2 from IFT trains by phosphorylating it, allowing reassembly of retrograde trains [11,74].

### 4.2. CDK

Another family of kinases involved in IFT regulation is CDK (cyclin-dependent kinases) and in particular CCRK in mammals (cell cycle-related kinase; a.k.a. CDK20) [159]. This kinase was shown to be involved in cancer, cancer metastatic suppression, hedgehog signaling and IFT regulation [183,184,185,186]. Here we focus on the effect of CCRK and its homologs (LF2 in *Chlamydomonas*, DYF-18 in *C. elegans*, Table 1) have on IFT. Knockout mutants of CCRK in mammals, *Chlamydomonas* and *C. elegans* have longer cilia with more length variability [160,161,175,186]. Furthermore, in mammalian cilia, accumulations of IFT components and shedding of vesicles from the tip have been observed [185,186]. Overall, the phenotype is similar to loss of RCK function (see above). IFT regulation by mammalian CCRK is dependent on another protein, Bromi (Broad-minded). Bromi has been suggested to form a complex with CCRK, promoting its stability [185,186,187]. Homologs of Bromi have not been reported in *Chlamydomonas* or in *C. elegans*. In *C. elegans*, the phosphorylation target of DYF-18 has not been identified, but in mammals and *Chlamydomonas* the RCKs CILK1 and LF4 are targets of CCRK and LF2, respectively [153,177,188]. Therefore, in these organisms, CCRK activates RCK by phosphorylation, which in turn can phosphorylate heterotrimeric kinesin-2. In mammals, pp5 (protein phosphatase 5) has been identified to play an additional role in this regulation network, by dephosphorylating CILK1 [177,188,189,190]. An interesting hypothesis is that the regulation network consisting of CCRK, pp5 and CILK1 is involved in regulating kinesin-II activity (Figure 5), such that it attaches to and drives anterograde IFT trains at the base, and detaches from trains at the tip. 

It is unclear what the actual effect of phosphorylation by CILK1 is on heterotrimeric kinesin-2 activity: does it regulate aspects of motor activity or attachment of the motor to IFT trains? Is the effect different than phosphorylation by other kinases (PKA and CAMKII, see below) [74,191]? Furthermore, it is unclear how the spatial distribution of CCRK, pp5 and CILK1 (activated and inactivated) would play a role. An additional interesting aspect is whether similar mechanisms are involved in the regulation of the two kinesin-2 motors involved in anterograde IFT in *C. elegans*, and if so, how. 

### 4.3. CaMK

Mammalian CaMKII (Ca^2+^/calmodulin-dependent protein kinase II) is a Ca^2+^-activated kinase [162,163] that regulates memory formation [192] by phosphorylating motor proteins [193] and cytoskeletal components [194]. In human dendrites, CaMKII regulates the interaction of KIF17 with the scaffolding protein Mint1. KIF17 and Mint1 together transport the NMDA-receptor subunit 2B, which is involved in learning and memory [195,196]. CaMKII acts as a molecular switch, by phosphorylating the C-terminal tail of KIF17. This disrupts the interaction between KIF17 and Mint1, releasing the cargo [193]. In *Chlamydomonas* motile cilia, the interaction between heterotrimeric kinesin-2 and IFT-B appears to be regulated in a similar way by CrCDPK1 (*Chlamydomonas reinhardtii* calcium-dependent protein kinases, a CaMKII homolog). CrCDPK1 phosphorylates the C-terminal tail of FLA8/KIF3B, thereby disrupting the interaction between kinesin-2 and IFT-B. This allows IFT trains to turn around and inactivated kinesin-2 to diffuse back to the base [22] (Figure 5). Together with CrPP1 (protein phosphatase 1) and CrPP6, CrCDPK1 regulates the phosphorylation state of FLA8/KIF3B, which in turn controls the flagellar entry of IFT trains and, as a consequence, flagellar length [74,197]. UNC-43, the *C. elegans* homolog of CaMKII, has not yet been reported to play a role in cilia. It is, however, involved in the locomotion of the whole animal [164,165].

### 4.4. CDKL

CDKL (Cyclin-Dependent Kinase-Like) family members are very similar to CDKs, apart from the fact that they do not seem to interact with cyclins [168]. CDKLs that have been shown to affect IFT are CDKL-5 in mammals, the closely related LF5 in *Chlamydomonas*, and CDKL-1 in *C. elegans* (which is more closely related to mammalian CDKL1-4). These kinases localize at the ciliary base or proximal segment, and in the case of mammals, also at the tip. The localization of *Chlamydomonas* LF5 and *C. elegans* CDKL-1 have been shown to depend on CDKF [168,169,170,198]. In *C. elegans* CDKL-1 knockouts, the length of the proximal segment is affected, while the overall structural integrity does not appear to be affected [168,169,198]. The role of CDKL in cilia in general remains an open question; it is unclear whether and how it affects ciliary structure.

### 4.5. Regulatory Proteins: Concluding Remarks

As shown above, different classes of kinases have been found to be involved in the regulation of IFT and ciliary structure in several model systems. The current picture might, however, not be complete. Many questions remain: are the networks of kinases and phosphatases uncovered so far complete? What functional properties of IFT components (e.g., motors) do these networks affect? Furthermore, RCK and CDK family members have been shown to play additional roles in regulating PTMs of the axonemal MTs, e.g., increasing acetylation of MTs resulting in their stabilization. This has been shown for MAK (RCK family) in mouse photoreceptors (which are highly specialized cilia) [199]. In addition, in the wing-shaped olfactory AWA cilia of *C. elegans*, it has been observed that MTs are more stable in the absence of DYF-18 (CDK family) [200]. This raises the question whether we have complete knowledge of the phosphorylation targets of these kinases and phosphatases. One would expect that not only heterotrimeric kinesin-2 is regulated by phosphorylation—as has been observed, but that also the other IFT motor proteins (IFT dynein, and, in some organisms, homodimeric kinesin-2) need to be regulated. Further studies will be required to identify these targets, which might employ techniques such as proximity-dependent biotinylation identification (BioID) [201] or affinity purification coupled with mass spectrometry [202,203], followed by mutagenesis studies.

## 5. Open Questions

Significant strides have been made in expanding our understanding of IFT regulation on a mechanistic level. Several important questions, however, remain to be addressed. 

Although the picture regarding the tubulin code is rapidly evolving, it is still largely unknown how MAPs, MIPs and PTMs decorate the axoneme, whether this distribution is heterogeneous and if that plays a role in regulating IFT. Furthermore, little is known regarding MT dynamics at the axoneme tip and the role played by MT-tip associated proteins in regulating this. 

A lot is understood about the action of kinesin-2 motors in driving anterograde IFT, although there are still some gaps in our understanding of collective motor action, autoinhibition and train interaction kinetics. At an enzymatic level, the role played by kinases in regulating kinesin-2 activity is slowly emerging, but the broader picture is still lacking. The interplay between heterotrimeric kinesin-2 and homodimeric kinesin-2 in regulating anterograde IFT is still unclear in many organisms. For IFT dynein, important new structural insights have been obtained, and we are starting to understand how the different (regulatory) subunits of the motor are involved in its autoinhibited state, when it is bound to anterograde IFT trains. A big open question is, however, how IFT dynein switches from the autoinhibited to an active state during anterograde-to-retrograde conversion of trains at the ciliary tip. Furthermore, while we know that IFT-dynein activity is tightly regulated, it is not clear what the regulatory players are that drive rapid redistribution of IFT dynein and other IFT components after external stimuli [32].

With the recent advances in cryo-EM techniques, our structural understanding of anterograde IFT-train assemblies (and its link with autoinhibited IFT-dynein) is improving rapidly, even at the level of how different IFT proteins interact with each other [204]. Future studies will likely provide a high-resolution picture of retrograde trains and binding interactions of IFT trains with IFT motors. Our knowledge on the anterograde-to-retrograde conversion of IFT trains at the cilia tip, and the assembly of anterograde IFT trains at the cilia base is slowly improving. Very little is known, however, about the exact location and timing of retrograde train disassembly at the base. 

Generally, the function of regulatory players of IFT have been studied by investigating ciliary structure and function in mutant organisms, where uncovering the actual role of the regulatory protein is often blurred by secondary and tertiary effects. Studies where the activity or interaction kinetics of proteins can be dynamically switched on and off, for example by optogenetics [205,206], chemical means [89], or auxin-inducible degron system [207,208], would significantly improve our understanding of the functional role of IFT proteins. Furthermore, recently it was found in *C. elegans* primary cilia that the steady state of IFT regulation can be promptly and substantially altered by providing external stimuli [32]. It is not clear yet whether such dynamic IFT-regulation mechanisms are also active in cilia of other species. Finally, it is important to keep in mind that ciliary structure, function and IFT mechanisms vary significantly between cilia and organisms. It might prove to be very helpful to explore beyond the typical ciliary model systems in order to obtain a broader picture of IFT regulation. 

## Figures and Tables

**Figure 1 cells-11-02737-f001:**
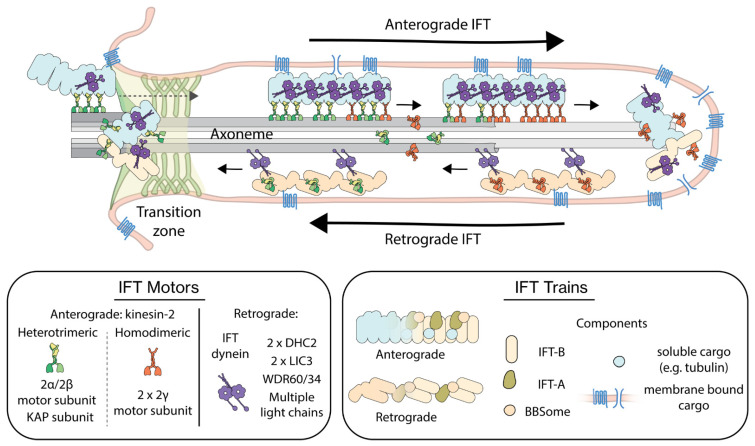
Illustration of intraflagellar transport (IFT) in primary cilia. Anterograde IFT trains assembled at the ciliary base are transported by kinesin-2 motors into the cilium, importing cargo proteins hitchhiking on the trains. At the ciliary tip, anterograde trains are converted to retrograde trains, which are driven back by IFT-dynein motors to the base. In the boxes, the composition of the different players of IFT have been highlighted. Heterotrimeric kinesin-2 is composed of two distinct motor subunits (2α/2β) and a non-motor KAP subunit, while homodimeric kinesin-2 is composed of two identical motor subunits (2γ). IFT dynein is a multi-protein complex consisting of two heavy chains (DHC2), two light intermediate chains (LIC3), two non-identical intermediate chains (WDR60/34) and several light-chain dimers. IFT trains are built up of three multiprotein-complex subunits, IFT-A, IFT-B and BBSome and carry cytosolic as well as membrane-bound cargo.

**Figure 2 cells-11-02737-f002:**
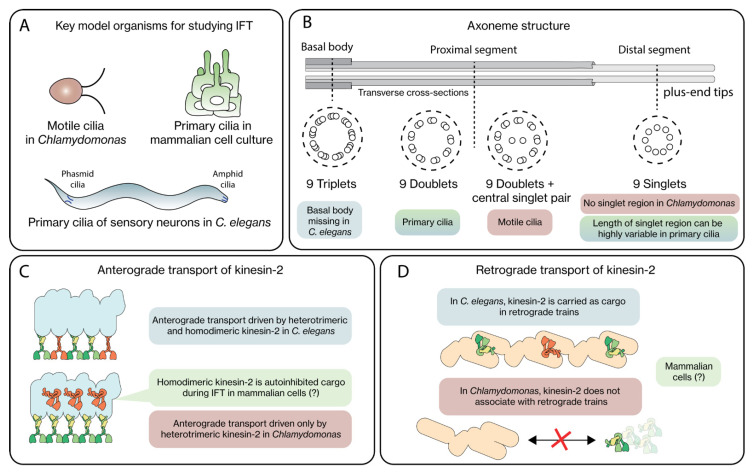
Differences in ciliary structure and IFT in key model organisms. (**A**) The motile cilia in *Chlamydomonas*, primary cilia of sensory neurons in *C. elegans* and the primary cilia of cultured mammalian cells are the most studied model organisms to study IFT. (**B**) The axoneme structure consists of 9 symmetrically arranged MT doublets, with their plus ends pointing outwards from the base to the proximal segment. In primary cilia, singlet MTs emanate from the MT doublets to form the distal segment. (**C**) In *C. elegans*, anterograde IFT is driven by heterotrimeric and homodimeric kinesin-2 motors, in contrast to *Chlamydomonas*, where heterotrimeric kinesin-2 is the sole driver of anterograde motion. In mammalian cells, the role of homodimeric kinesin-2 is still unclear. (**D**) In *C. elegans*, kinesin-2 motors link with retrograde IFT trains to be transported back towards the base, in contrast to *Chlamydomonas*, where the motors diffuse back. It is still unknown what scenario mammalian cells use.

**Figure 3 cells-11-02737-f003:**
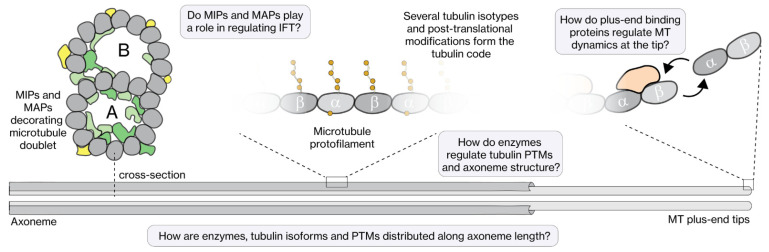
The axoneme track and its regulators. The MTs (A-tubule and B-tubule) that make up the axoneme structure are heavily decorated with MIPs and MAPs, which potentially play a role in regulating IFT. The tubulin lattice itself is composed of several tubulin isotypes that have several post-translational modifications (acetylation, glutamylation, detyrosination, etc.), which form the tubulin code. How regulating enzymes modulate the tubulin code and axoneme structure, how the tubulin code influences IFT, and how dynamics of the MT tips are regulated, are still open questions in the field.

**Figure 4 cells-11-02737-f004:**
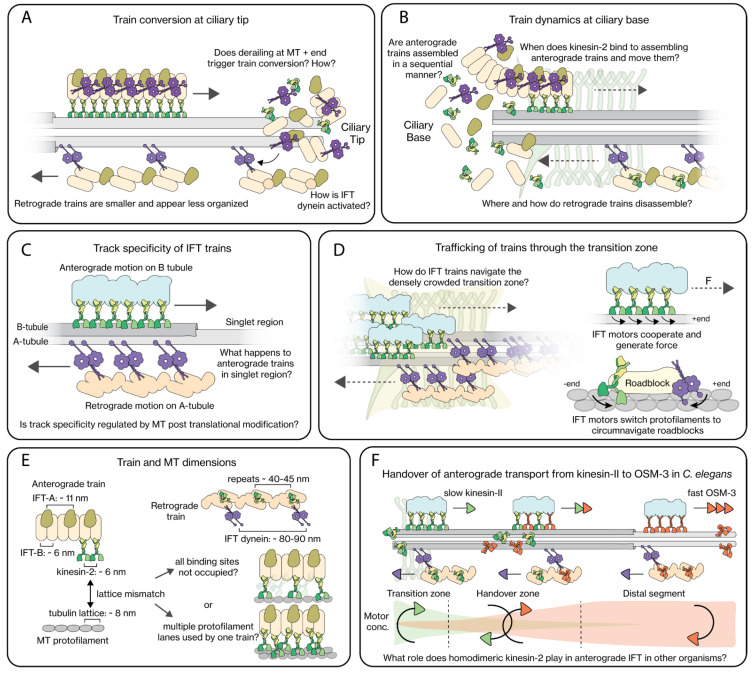
Mechanistic insights into different aspects of IFT. (**A**) At the ciliary tip, highly structured anterograde trains get converted to smaller and less ordered retrograde trains. It is still unclear how this conversion is orchestrated, and how IFT dynein switches from an autoinhibited to the active state. (**B**) At the ciliary base, large anterograde trains are assembled, possibly in a sequential manner, with activated kinesin-2 motors dragging them into the cilium after they are assembled. Little is understood of retrograde train disassembly at the base. (**C**) In *Chlamydomonas*, anterograde trains move along the B-tubule while retrograde trains move along the A-tubule of MT doublets. (**D**) IFT motors have to navigate IFT trains through a densely crowded transition zone. The IFT motors can generate force and are capable of circumnavigating roadblocks by switching protofilament lanes, which likely allow them to handle this. (**E**) Anterograde trains have kinesin-2 binding sites every ~6 nm, while MT protofilaments have tubulin repeats every 8 nm. This implies that, at any given time, not all kinesin-2 motors engage with the MT lattice and/or kinesin-2 motors use multiple protofilament lanes to transport an anterograde train. IFT-dynein motors bind ~80–90 nm apart from each other on retrograde trains. (**F**) In *C. elegans*, anterograde trains are driven across the transition zone by the slower heterotrimeric kinesin-2 (kinesin-II), after which trains are gradually handed over to the faster homodimeric kinesin-2 (OSM-3), which carry the trains to the ciliary tip. The role played by homodimeric kinesin-2 in other organisms is not well understood.

**Figure 5 cells-11-02737-f005:**
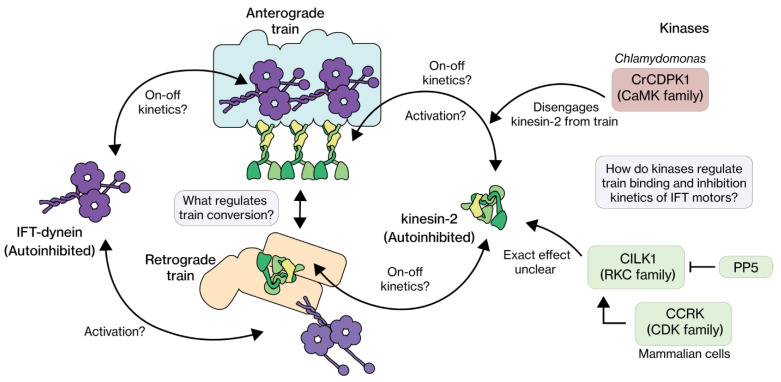
The IFT train binding and autoinhibition kinetics of IFT motors. Kinesin-2 motors are active when coupled to anterograde trains, but switch to an autoinhibited state upon unbinding, linking as passive cargo on retrograde trains or moving diffusively. IFT dynein engages with anterograde trains in an autoinhibited state and become active upon binding to retrograde trains, at the ciliary tip. Kinases and other regulators are likely involved in regulating the train (un)binding and autoinhibition kinetics, but the picture is still quite unclear, especially for IFT dynein.

**Table 1 cells-11-02737-t001:** Family names and names of kinases involved in IFT regulation.

	Mammals	*C. elegans*	*Chlamydomonas*
RCK	CILK1 and MRK [152,153]	-	-
	MAK [154,155]	Dyf-5 [156]	-
	MOK [157]	-	LF4 [158]
CDK	CCRK/CDK20/p42 [159]	Dyf-18 [160]	LF2 [161]
CaMK	CaMKII [162,163]	UNC-43 [164,165]	CrCDPK1 [166]
CDKL	CDKL5 [167,168]	CDKL-1 [169]	LF5 [170]

## Data Availability

Not applicable.

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
