# Peer review of "Mechanisms of Regulation in Intraflagellar Transport"

_cells, 2022, doi:10.3390/cells11172737_

Round 1

Reviewer 1 Report

The authors provided accurate and thorough discussions of the current findings of the IFT regulation. This reviewer enthusiastically recommends the publication of the manuscript in cells after minor corrections.

Minor points:

1.     The authors can mention the enrichment of tubulin polyglutamylation in the B-tubule (Lechtreck and Geimer 2000 PMID: 11056523) in the “The tubulin code” or “Track specificity of IFT trains” sections.

2.     The authors can mention polyglycylation in “The tubulin code” section.

3.     The authors can cite Kubo et al. 2016 (PMID: 27068536) as another reference for the anterograde IFT of tubulin. (Line 357)

4.     The authors can cite Jiang et al. 2022 (PMID: 35969738) regarding the DYF-5-dependent ciliogenesis. (Line 602)

Line 73: Chlamydomonas rheinhardtii -> reinhardtii

Line 150: detyronisation -> detyrosination

Line 160: deglutamination -> deglutamylation

Line 160: Poly-glutamination-> poly-glutamylation

Line 172: how dynamics of the MT tips is regulated -> are regulated

Line 259: apart fromn ->  apart from

Line 331: 2.2µm/s -> 2.2 µm/s

Author Response

We thank the reviewer for their very positive comments and remarks to make our manuscript more accurate. We have implemented all the corrections and suggestions made by the reviewer.

Reviewer 2 Report

The review by Mul et al is a timely discussion of mechanisms of regulation of intraflagellar transport (IFT). The authors focus on three different aspects of IFT regulation: role of axoneme in IFT, role of motors and motor-train interactions in the regulation of IFT and the role of kinases in the regulation of motors. The review is well researched and nicely written. The figures fit in the narrative well. The authors also suggest important directions of future research.

I have a few major comments that might be relevant for discussion of this important topic in cilia cell biology.

First, the regulation of IFT turnaround in cilia tips can include discussion about accumulation of Gli, Sufu, Kif7 etc., especially considering recent work on Kif7 and Gli.

Second, the regulation of anterograde trains and IFT-A can include discussion of the adapter TULP3 with relation to cargo trafficking into cilia.

Minor points:

-Line 54-55: References to GPCR trafficking (16-19): Please revise. The BBSome is now generally thought to regulate exit of cargos from cilia.

-Line 615-616: IFT-A mutants don’t have long cilia like in RCK knockouts.

Author Response

We thank the reviewer for their positive comments. We have implemented the two major suggestions and corrected the two minor points.
